# Low Arousal Threshold Estimation Predicts Failure of Mandibular Advancement Devices in Obstructive Sleep Apnea Syndrome

**DOI:** 10.3390/diagnostics12102548

**Published:** 2022-10-20

**Authors:** Caterina Antonaglia, Gabriele Vidoni, Luca Contardo, Fabiola Giudici, Francesco Salton, Barbara Ruaro, Marco Confalonieri, Martina Caneva

**Affiliations:** 1Pulmonology Department, University Hospital of Cattinara, 34149 Trieste, Italy; 2Department of Medicine, Surgery and Health Sciences, School of Dentistry, University of Trieste, 34127 Trieste, Italy; 3Department of Biostatistics and Epidemiology, Gustave Roussy, Paris-Saclay University, 91190 Gif-sur-Yvette, France; 4Department of Medicine, Surgery and Health Sciences, University of Trieste, 34127 Trieste, Italy

**Keywords:** mandibular advancement devices (MADs), low arousal threshold (low ArTH), obstructive sleep apnea syndrome (OSAS)

## Abstract

Introduction: The treatment of choice for obstructive sleep apnea syndrome (OSAS) is continuous positive airway pressure (CPAP). However, CPAP is usually poorly tolerated and mandibular advancement devices (MADs) are an alternative innovative therapeutic approach. Uncertainty still remains as to the most suitable candidates for MAD. Herein, it is hypothesized that the presence of low arousal threshold (low ArTH) could be predictive of MAD treatment failure. Methods: A total of 32 consecutive patients, with OSAS of any severity, who preferred an alternate therapy to CPAP, were treated with a tailored MAD aimed at obtaining 50% of their maximal mandibular advancement. Treatment response after 6 months of therapy was defined as AHI < 5 events per hour or a reduction of AHI ≥ 50% from baseline. Low ArTH was predicted based on the following polysomnography features, as previously shown by Edwards et al.: an AHI of 82.5% and a hypopnea fraction of total respiratory events of >58.3%. Results: There were 25 (78.1%) responders (*p*-value < 0.01) at 6 months. Thirteen patients (40.6%) in the non-severe group reached AHI lower than 5 events per hour. MAD treatment significantly reduced the median AHI in all patients from a median value of 22.5 to 6.5 (74.7% of reduction, *p*-value < 0.001). The mandibular advancement device reduced AHI, whatever the disease severity. A significant higher reduction of Delta AHI, after 6 months of treatment, was found for patients without low ArTH. Conclusions: Low ArTH at baseline was associated with a poorer response to MAD treatment and a lower AHI reduction at 6 months. A non-invasive assessment of Low ArTH can be performed through the Edwards’ score, which could help to identify an endotype with a lower predicted response to oral appliances in a clinical setting.

## 1. Introduction

Obstructive sleep apnea syndrome (OSAS) is the most frequent breathing-related sleep disorder where upper airway collapse during nighttime leads to a reduction (hypopnea) or interruption (apnea) of the airflow [1]. Obstructive events are followed by phasic oxyhemoglobin desaturations, sympathetic hyperactivation, and sleep fragmentation. These events are the main underlying causes that make OSAS a risk factor for cardiovascular diseases, hypertension, daytime sleepiness, work and road-related accidents, and consequent worsening of life quality [1,2]. Whilst about 30% of patients have anatomical predisposing features, i.e., obesity, retrognathia, laxity of the soft palate, and macroglossia, recently it has been reported that in other 70% of the cases there are non-anatomical factors also involved in the disease pathogenesis [3,4] (instability of ventilatory control, also known as high loop gain; neuromuscular inefficiency of the dilator muscles of the upper airways and increased propensity for nocturnal awakenings due to respiratory stimuli, or a reduced awakening threshold, also known as low arousal threshold (low ArTH), which seems to fit up to 30–50% of cases [5]. The upper airway collapse that occurs in patients affected by OSAS during sleep increases carbon dioxide levels and, consequently, ventilatory drive. However, if there is a low ArTH, respiratory events terminate earlier, meaning that the ventilatory drive has insufficient time to build up and restore pharyngeal patency without arousal. ArTH refers to the neuromuscular mechanical pressure present at the end of an apnea–hypopnea event, responsible for awakening from sleep–arousal and can only be quantified invasively by an epiglottic or esophageal pressure catheter [5]. A recent study by Edwards et al. reported that low AT could be estimated non-invasively through the following clinical score which attributes a point for each criterion met between an AHI ≤ 30 events/h, a nadir SpO_2_ ≥ 82.5%, and a hypopnea fraction of total respiratory events of > 58.3%. A score of ≥2 predicts a low AT in OSAS patients, with high sensitivity and specificity (80.4% and 88% respectively) [5]. Other authors support the idea that it is possible to identify which of the non-anatomical factors contributes most to the pathogenesis of OSAS based on the characteristics of sleep study tracing [6].

To date, the use of continuous positive airway pressure (CPAP), through a nasal or oronasal mask, is the only treatment documented as being effective in suppressing respiratory disturbances during sleep and improving clinical manifestations. Its use for more than six hours has decreased sleepiness, improved daily functioning, and restored memory to normal levels [7,8]. However, a number of factors make the overall adherence to CPAP unsatisfactory, including physical and psychological discomfort with the device [9]. This has prompted new tailored therapeutic approaches which have been the object of recent increasing interest [10]. Oral appliances (OA) have been shown to be a good alternative or supplement to CPAP. OAs are designed to improve upper airway configuration and prevent collapse by altering the position of the jaw and tongue. The most common mechanism of action is to keep the lower jaw in a more anterior position. Indeed, they have been indicated by the latest American Academy of Sleep Medicine (AASM) guidelines for all patients who are either intolerant to CPAP or prefer alternate therapy, as well as a first-line treatment for primary snoring without OSA [11]. The most commonly used OAs are mandibular advancement devices (MADs), reported as normalizing the apnea-hypopnea index (AHI) between 36% and 70% of patients, reducing upper airway collapsibility upon protrusion of the lower jaw and tongue [11]. Based on the definition of 4 h/night, literature data reports an adherence of up to 76% in patients treated with MADs alone versus 43% in patients treated with CPAP alone [12,13]. Although the most recent AASM guidelines recommend the use of MADs, they do not address patient eligibility, due to the lack of data on the identification of the best candidates in terms of predicted reduction of both symptoms and risk for future impaired health [14]. The most commonly used definitions of response are (1) treatment AHI < 5/h or complete resolution of OSA (definition 1), (2) treatment AHI < 10/h and ≥50% reduction in AHI from baseline (definition 2), and (3) ≥50% reduction in AHI from baseline (definition 3) [15].

The efficacy of a specific, tailor-made, mandibular advancement device, named “Silensor” (Erkodent Eirch Kopp GmbH, Pfalzgrafenweiler, Germany) was assessed in OSAS patients with mild to severe disease. It was hypothesized that the presence of a non-anatomical pathophysiological factor, like a low arousal threshold, may well explain the poor response to MAD therapy and that it could be used to predict treatment failure.

## 2. Methods

### 2.1. Study Design, Setting, and Participants

The study was performed between July 2015 and December 2020 in the Pulmonology Department and the Dental Unit of the University Hospital of Trieste (Italy). A total of 32 patients with either mild, moderate, or severe OSAS. Patients with mild OSA had clinical and anatomical features such that MAD was considered the first treatment choice. All severe and 10 moderate patients had a trial of CPAP first (18 patients), but they then refused this treatment. Patients preferred not to use CPAP due to their personal preference despite medical counseling, so we offered them evaluation for MADs. The study was carried out according to the declaration of Helsinki after approval by the local Ethical committee. Each patient provided written informed consent for the study. The diagnosis of OSAS was made by the use of a home sleep apnea test (HSAT) SOMNOlab2 (Weinmann, Hamburg, Germany). The recordings included the nasal cannula channel, pressure sensor channel, and arterial oxygen saturation, thoracic and abdominal bands were also monitored. All recordings were scored visually by one experienced rater.

Apnea was defined as oronasal flow cessation for more than 10 s. Hypopnea was defined as a 50% reduction in oronasal flow, followed by a more than 3% decrease in SaO2 Based on the polysomnography results, OSA was defined as an apnea-hypopnea index (AHI) > 5 per h, of which ≥80% were obstructive. Mild-to-moderate OSA was defined as AHI > 5 per h and AHI ≤ 30 per h. Severe OSA was defined as AHI > 30 per h [16]).

Inclusion criteria were: (a) diagnosis of OSAS dated less than 30 days prior to inclusion; (b) the patient not being on any specific OSAS therapy; (c) the absence of grade III tooth mobility; (d) the absence of any acute temporomandibular disorders; (d) the presence of at least 20 teeth. Exclusion criteria were any clinical or historical element that according to the clinician’s discretion, might have put patients at substantial risk or have compromised treatment outcomes, due to poor compliance.

The MAD used in this study was Silensor or Silensor-sl (Erkodent Eirch Kopp GmbH, Pfalzgrafenweiler, Germany). It is an adjustable device made of two transparent splints: one for the upper jaw and one for the lower jaw. These are held in the predetermined position by two laterally fixed connectors that can be modified to increase or decrease the degree of protrusion. The device was tailored to each patient by obtaining alginate impressions of the jaws to maintain 50% of the maximal mandibular advancement, as assessed by a protrusion gauge. The patients were taught how to insert and remove the device, as well as its hygiene and maintenance.

The study baseline was defined as the date of the first MAD insertion. The follow-up visits were scheduled at 1, 2, and 6 months from baseline. Patient compliance, symptom reduction, and the degree of mandibular advancement were assessed at each visit measured as mm but also as % of the maximal protrusion. A second nocturnal cardiorespiratory monitoring was performed with the SOMNOlab 2 device after 6 months and the body mass index (BMI) of all patients was recorded. Treatment response was defined as an AHI of < 5 events per hour or (b) a ≥ 50% reduction in AHI from baseline [15]. A therapeutic alternative, e.g., CPAP, was offered to non-responders, as assessed by the follow-up nocturnal cardiorespiratory monitoring. Patients who took drugs capable of modifying the low AT were excluded from the study.

The nocturnal cardiorespiratory monitoring data provided information on the presence of a low ArTH using the Edwards’ score [5]. One point was attributed to an AHI lower than 30 events per hour, a nadir SpO_2_ higher than 82.5% and a hypopnea fraction higher than 58.3%. Patients scoring ≥ 2 were defined as having a low ArTH.

### 2.2. Statistical Methods

The results on continuous variables were reported either in terms of mean ± standard deviation (SD) or as median and range (minimum and maximum). Whilst absolute frequencies and percentage values were used for categorical variables. The Shapiro-Wilk test was applied to continuous variables to assess the distribution normality. A comparison of numerical parameters, before and after treatment, was made by Student’s *t*-test for paired data or the related non-parametric Wilcoxon test. Whilst categorical parameters were compared by the McNemar test or the Stuart-Maxwell homogeneity test when there were more than two categories per variable. The relationship between absolute (delta) and percentage reduction of AHI compared to the baseline variable and the parameters of interest was assessed by the Mann-Whitney U test or the Kruskal-Wallis H test (depending on the number of groups to be compared). The Spearman linear correlation coefficient was calculated for the variable mandibular advancement in mm and BMI. Pearson’s chi-squared test (or Fisher’s exact test, depending on the number of groups to be compared) was used to identify any correlations between categorical variables. Data analysis was performed with R software (ver. 4.0.2, 2002) and the statistical significance level was set at a *p*-value < 0.05. All tests were performed as two-tailed.

## 3. Results

Between July 2015 and December 2020, 34 patients were evaluated for inclusion in the study. A total of 2/34 did not meet inclusion criteria and were excluded. One patient was edentulous, whilst the other had a class III malocclusion with an anterior cross-bite and limited protrusion capacity.

Table 1 reports the baseline characteristics of the study population. Most patients were either in the Mallampati class III or IV (37.5% and 40.6%, respectively). The mean mandibular advancement was 4.1 ± 1.2 mm, corresponding to 50% of the maximal protrusion for each patient.

A total of 8/32 (25%) was classified as having severe OSAS, 17/32 (53.1%) moderate, and 7/32 (21.9%) mild (Table 2). All patients were compliant with MAD for more than 4 h per night.

At 6 months, all patients referred a reduction of choking, apnea reported by the bed partner, daytime sleepiness, and loud snoring, which were the major symptoms reported to our patients.

The following criteria were used to evaluate treatment response: (a) an AHI of < 5 events/h or (b) an AHI reduction from baseline of ≥ 50% [17]. A total of 7/32 patients (21.9%) were non-responders, whilst 25/32 (78.1%) were responders (Table 3).

Thirteen patients (40.6%) fell within the physiologic range of obstructive sleep apnea (AHI < 5), however none of them had severe OSAS at baseline. Noteworthy is the fact that the response rate was not significant for mild (57.1% vs. 42.9%, *p*-value = 0.73) and severe (87.5% vs. 12.5%, *p*-value = 0.11) patients but it was for moderate patients (82.6% vs. 17.4%, *p*-value = 0.03).

AHI reduced from a median value of 22.5 (7.6–76.6) to 6.5 (0–23.6), 17.0 between-group difference, corresponding to a 74.7% reduction, a *p*-value of <0.001. There was a statically significant reduction in the AHI in the supine position, from a median value of 32.6 (13.1–91.3) to 8.7 (0.0–47.9), with a *p*-value < 0.001. Whilst the non-supine position AHI reduced from 5.5 (0–62.4) to 2.2 (0–18.2), *p*-value = 0.03 (Table 4).

For severe OSA patients the median AHI reduction was 36.6 (19.3–66.3), for moderate patients it was 16.8 (−4.7–24.7), and for mild ones it was 6.7 (2.0–12.3). A greater variation of delta AHI was observed to be related to a higher value of baseline AHI (*p* < 0.001) (Figure 1).

The mean BMI was 26.2 ± 4.4 at baseline and remained stable at 6 months. There was no statistically significant correlation between the BMI and the AHI reduction, nor between the grade (expressed in millimeters) of mandibular advancement and the AHI reduction. Similarly, there was no statistically significant difference in the AHI reduction in patients who had a mandibular advancement of more than 4 mm compared to those with a mandibular advancement of less than 4 mm.

A trend towards significance in the AHI reduction was observed between patients with Mallampati class 3 versus Mallampati class 4 (84% vs. 63.1%, *p*-value = 0.12).

Edward’s score for ArTH was available for 23/32 patients. A significantly higher reduction of Delta AHI, after 6 months of treatment, was found for patients without low ArTH (delta AHI: 20.2 vs. 12.2 *p* = 0.03) (Figure 2 and Figure 3).

However, looking at treatment response defined as AHI < 5/h, we observe a higher percentage of responding patients in the group of patients with low ArTH (58% vs. 27%, *p* = 0.13), but as is well known in literature, even in our patients a low ArTH is associated with less severe diseases (Table 5).

## 4. Discussion

We showed the efficacy of individualized MAD in patients with OSAS at all levels of severity and tested the hypothesis that the presence of a pathophysiological trait, such as low ArTH, can explain and help predict treatment failure. Although MADs were originally designed to treat primary snoring, several recent studies have demonstrated their efficacy in reducing both AHI and daytime symptoms also in patients with OSAS [18]. The most recent AASM guidelines recommend tailored MAD over non-personalized ones in OSAS patients who cannot tolerate CPAP or prefer alternate therapy, but evidence to guide patient eligibility is scanty [14].

On the basis of literature data, patients were defined as responders if they either fell within the normal AHI range (complete response) or had an AHI reduction of more than, or equal to, 50% from baseline [17]. When analyzing these two criteria independently, 40.6% of the patients obtained a complete response from the personalized MAD, none of them had been classified as severe at baseline. This is in line with previous data from several studies that reported a complete response in 29% to 71% of cases [19]. The overall response rate in this study at 6 months was as high as 78%.

However, taking into account the different disease severity categories, i.e., mild, moderate, and severe, the response rate was only statistically significant in the moderate category. This might be partially due to the small sample size, as a trend toward significance was observed in severe OSAS patients.

The median AHI reduction in all patients was 74%, which was statistically significant with a high confidence interval. Moreover, the higher the baseline AHI, the greater the AHI variation after 6 months of treatment. In support of these results, all patients reported a reduction in their symptoms such as choking, apneas reported by the bed partner, daytime sleepiness, and loud snoring, as already reported by other Authors [12].

A recent review [20] identified the features of responders to MAD therapy: younger age, female sex, lower body mass index, smaller neck circumference, lower apnea-hypopnea index, a retracted maxilla and mandible, narrower airways, and shorter soft palate than non-responders. Polysomnographic variables between responders and non-responders are also reported in the literature, e.g., responders have a less severe desaturation index and higher minimum arterial oxygen saturation; moreover, it is known that MADs are more effective in patients with positional OSA than those with non-positional OSA.

It has been demonstrated that the position and length of the mandible, in association with the tongue and soft palate area, are some of the most important anatomical features related to OSAS [21]. A total of 78.1% of the patients fell into the Mallampati class 3 or 4 and there was a trend toward a correlation between the Mallampati class and the AHI reduction. All patients have advanced 50% of their maximal protrusion, finding no correlation between the degree of mandibular advancement and AHI reduction, nor it was when comparing the results of patients who underwent mandibular advancement higher or lower than 4 mm. This is in agreement with literature data, where no studies identified a protrusion cutoff that correlated with a better response, despite the maximum achievable protrusion having been listed as an important predictor of treatment success [18].

A recent meta-analysis suggested that certain phenotypes or anthropometric characteristics may help predict clinical response to MADs [19]. In addition, data from the literature support that recognizing the presence of a nonanatomic pathophysiologic trait that predisposes to OSAS, such as high critical pressure, low arousal threshold, high loop gain, or low muscle responsiveness, could guide the choice of the best treatment for each patient [2]. For example, patients who require higher therapeutic CPAP pressure usually have a lower chance of response to MAD, as this reflects a higher baseline critical pressure [21] Edwards et al. found that loop gain at baseline was lower in responders than in non-responders to MAD therapy [22]. The most commonly found of these pathophysiologic factors is low ArTH, which is involved in approximately 30–50% of cases [19]. However, data regarding its possible predictive role in response to MAD are scarce [20].

Although MAD therapy in patients with OSA is more effective in patients with mild and positional syndrome, in our study we have shown how the efficacy of MAD in terms of reducing AHI is influenced by the presence of low ArTH. Indeed, in patients with low ArTH, MAD is less effective (in terms of delta AHI) because low ArTh contributes to the occurrence of obstructive events regardless of the severity of the syndrome and the anatomical alteration corrected by MAD.

In fact, low ArTH is believed to contribute to the pathology of OSAS, as repeated awakenings result in destabilizing effects [23], such as:(1)The absence of sufficient time for the respiratory drive to recruit the pharyngeal muscles and reopen the airways before arousal;(2)Reduced partial pressure of carbon dioxide, which promotes dynamic ventilatory instability, contributing to the perpetuation of respiratory events;(3)Fragmentation of sleep, which does not allow the individual to achieve slow wave sleep (i.e., to stabilize sleep).

In the literature, there was a strong correlation of ArTH with AHI, minimum peripheral oxygen saturation (SpO_2_), and the fraction or percentage of hypopneas compared to total respiratory events, which were therefore found to be strong predictors of a low arousal threshold.

Patients with low ArTH have less severe disease and less profound desaturation because they easily reach the arousal threshold before experiencing a severe desaturation or obstructive event [5].

In our study patients with low ArTH had a non-anatomic pathophysiological factor predictive of failure of MAD therapy.

A recent work by Edwards et al. [5] showed how, unlike the other pathophysiological mechanisms of OSAS, such as loop gain and neuromuscular inefficiency, low arousal threshold (low ArTH) is the only one that can be estimated non-invasively. The Edwards score does not use EEG and is not well validated in literature, but the quantifying arousal threshold in a laboratory is very expensive and complicated and many authors use this score to predict arousal [24]. This is a limit of our study, and it might be interesting to repeat the same study in the laboratory with complete polysomnography to validate our results.

The present study has some limitations, such as the small sample size, therefore it is strongly recommended to carry out randomized clinical trials on larger populations, in order to provide more definitive data for or against the findings here described. Another limitation is the use of only one study device, which was dictated by the experimental setting. However, it is likely results of the above study could be generalized to every MAD, which is something that would merit further development in the context of a randomized clinical trial. Moreover, although it has been investigated the predictive potential of low ArTH on MAD response, it was not evaluated whether, and to what extent, the same endotype would have predicted a lower response to other therapeutic approaches like CPAP, as already hypothesized by other studies. The main limitation of our study is the use of a home sleep study without EEG, so we tried to do an expert scoring of polygraphic tests to understand the prevalent pathophysiological trait of our patients. Our study is supported by the evidence in the literature that a specific cardio-respiratory monitoring pattern predicts the presence of a non-anatomical predisposing factor for OSA for that patient as Bosi et al. suggested [6].

Lastly, we only investigated low ArTH as it is the most frequent non-anatomical factor involved in the pathogenesis of OSAS, even if we are an area of the fact that it is not the only one. Hopefully, further studies focusing on different anatomical and pathophysiological features will lead to a more complex personalized approach, able to predict treatment response in every single patient.

In conclusion, it has been observed that a personalized MAD was an effective treatment option for OSAS and allowed for a significant AHI reduction, whatever the disease severity and there was a predominant clinical response in moderate patients. The Edwards’ score provided an easy evaluation of the presence of a low ArTH endotype, which was associated with a lower response to MAD treatment.

## Figures and Tables

**Figure 1 diagnostics-12-02548-f001:**
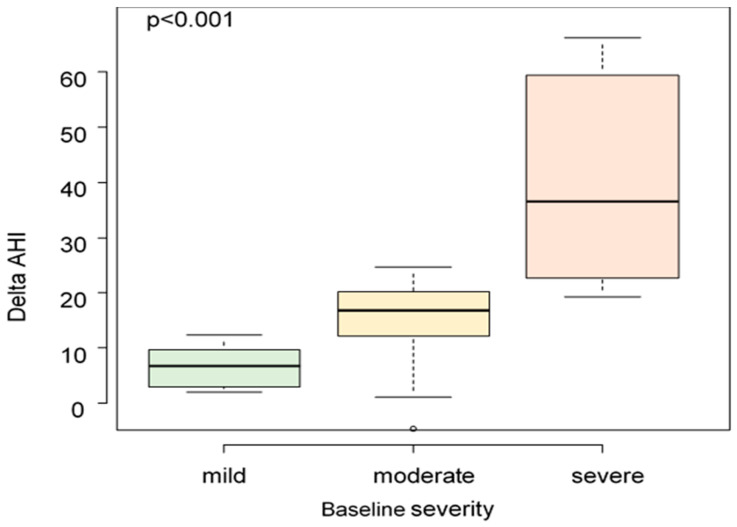
AHI variation at different baseline severities (*p*-value < 0.001).

**Figure 2 diagnostics-12-02548-f002:**
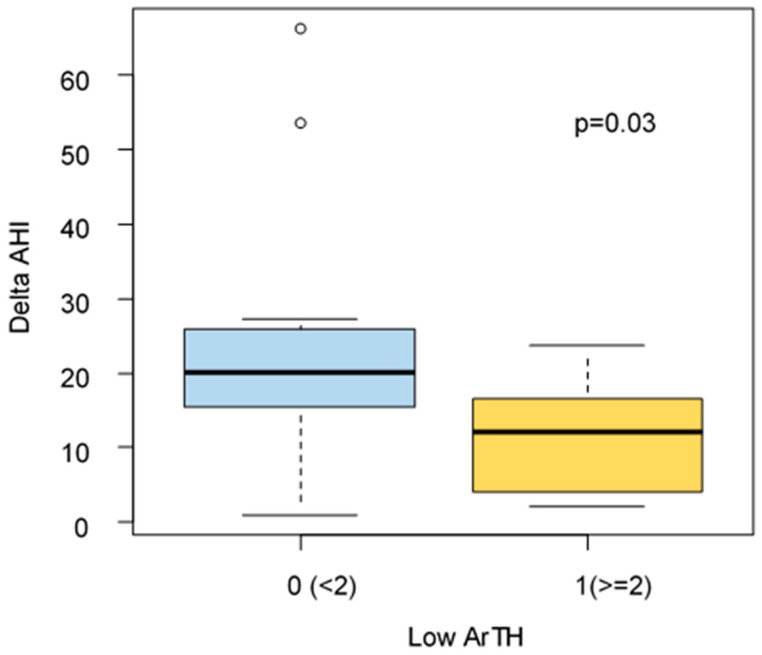
AHI variation related to low ArTH (*p*-value = 0.03).

**Figure 3 diagnostics-12-02548-f003:**
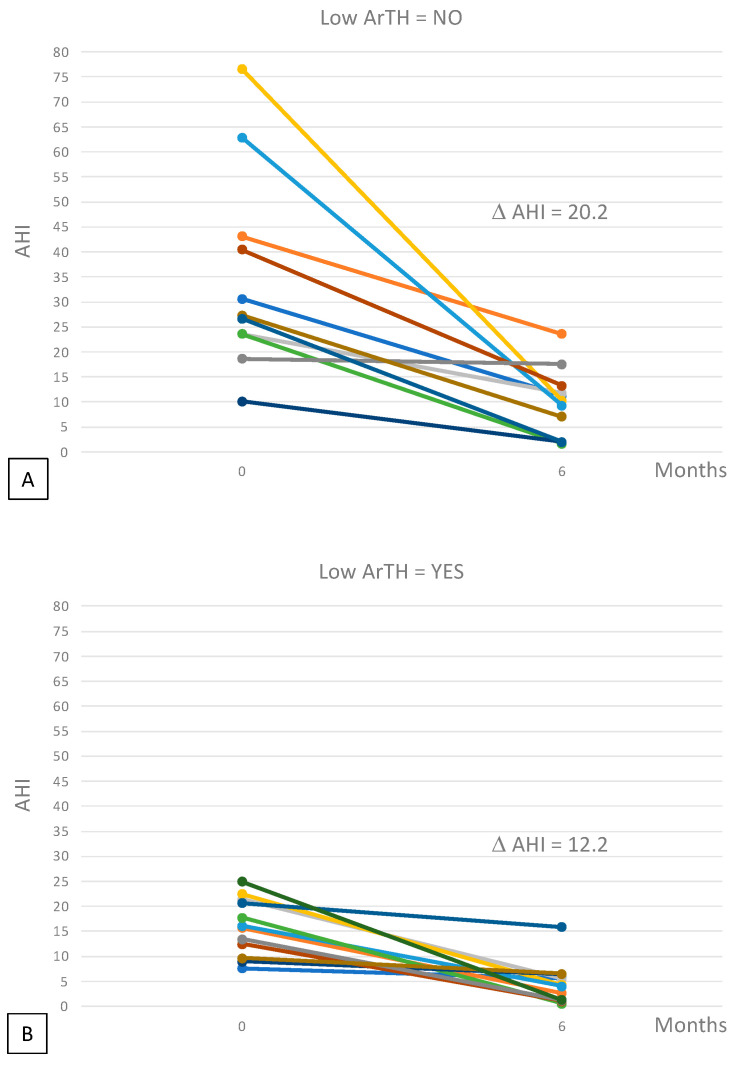
Spaghetti plots reporting the individual data (different color for each patients) of the variation of AHI among patients without low ArTH (panel (**A**)) and with low ArTH (panel (**B**)). D AHI was 20.2 vs. 12.2, respectively; *p* = 0.03.

**Table 1 diagnostics-12-02548-t001:** Baseline Patient Characteristics.

Variable	Patients (n = 32)
Age (years)	
Mean (St. Dev.)	57 (10)
Body Mass Index (BMI) Baseline	
Mean (St. Dev.)	26.2 (4.4)
Gender (n,%)	
Women	8 (25%)
Men	24 (75%)
Arterial Hypertension (n,%)	
Presence	14 (43.8%)
Cardiovascular Disease (n,%)	
Presence	5 (15.6%)
Diabetes Mellitus (n,%)	
Presence	2 (6.3%)
Mallampati Score (n,%) (1 NA)	
1	1 (3.2%)
2	5 (16.1%)
3	12 (37.5%)
4	13 (40.6%)
Dental occlusion (n,%) (2 NA)	
1	13 (43.3%)
2	16 (53.3%)
3	1 (3.3%)
mm MA (50%) (3 NA)	
Mean (St. Dev.)	4.1 (1.2)

**Table 2 diagnostics-12-02548-t002:** Baseline nocturnal cardiorespiratory monitoring data.

Variable	Value
AHI Baseline	
Median (Min–Max)	22.5 (7.6–76.6)
Severity at Baseline (N,%)	
Mild	7 (21.9%)
Moderate	17 (53.1%)
Severe	8 (25.0%)
ODI/h Baseline (1 NA)	
Median (Min–Max)	15.6 (1.5–74.4)
Sato2 min Baseline (4 NA)	
Median (Min–Max)	82.5 (59.0–91.0)
T < 90% Baseline (2 NA)	
Median (Min–Max)	1.6 (0.0–87.3)
Low ArTH (0 se < 2; 1 se ≥ 2) (9 NA)	
0	11 (47.8%)
1	12 (52.2%)

**Table 3 diagnostics-12-02548-t003:** Patients defined as responders, based on the definitions (AHI < 5/h and ≥50% reduction from baseline) or non-responders considering severity at baseline.

Classification	All	Mild	Moderate	Severe
**AHI < 5**	13 (40.6%)	4 (57.1%)	9 (52.9%)	0 (0.0%)
**≥50% AHI reduction**	12 (37.5%)	0 (0.0%)	5 (29.4%)	7 (87.5%)
**Responders**	25 (78.1%)	4 (57.1%)	14 (82.3%)	7 (87.5%)
		*p*-value 0.73	*p*-value 0.03	*p*-value 0.11
**Non-responders**	7 (21.9%)	3 (42.9%)	3 (17.6%)	1 (12.5%)
**Total**	32	7	17	8

**Table 4 diagnostics-12-02548-t004:** Polygraphic variable before and after treatment.

Variable	Baseline	After Treatment	*p*-Value
Total AHI (N = 32 pt)			
Median (Min–Max)	22.5 (7.6–76.6)	6.5 (0–23.6)	<0.001
Severity (N,%)			
Mild	7 (21.9%)	13 (40.6%)	
Moderate	17 (53.1%)	15 (46.9%)	<0.001
Severe	8 (25.0%)	4 (12.5%)	
ODI/h (N = 30 pt)			
Median (Min–Max)	15.6 (1.5–74.4)	3.4 (0.2–32.6)	<0.001
Sat O2 min (N = 26 pt)			
Median (Min–Max)	82.5 (59.0–91.0)	85.0 (59.0–91.0)	0.19
T < 90% (2 NA)			
Median (Min–Max)	1.6 (0.0–87.3)	0.55 (0–71.4)	0.03
Supine (N = 19 pt)			
Median (Min–Max)	32.6 (13.1–91.3)	8.7 (0.0–47.9)	<0.001
Non-Supine (N = 19 pt)			
Median (Min–Max)	5.5 (0–62.4)	2.2 (0–18.2)	0.03

**Table 5 diagnostics-12-02548-t005:** Patients’ responders (for one of the three definitions: AHI < 5/h, AHI < 10/h plus > 50% reduction from baseline, or ≥50% reduction from baseline) or non-responders with or without low ArTH.

Classification	Without Low ArTH	With Low ArTH
**AHI < 5**	3 (27.3%)	7 (58.3%)
**≥50% AHI reduction**	6 (54.5%)	1 (8.3%)
**Responders**	9 (81.8%)	8 (66.7%)
**Non-responders**	2 (18.2%)	4 (33.3%)
**Total**	11	12

## Data Availability

Deidentified participant data will be made available upon motivated request to the Corresponding Author.

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
