# Peer review of "Low Arousal Threshold Estimation Predicts Failure of Mandibular Advancement Devices in Obstructive Sleep Apnea Syndrome"

_diagnostics, 2022, doi:10.3390/diagnostics12102548_

Round 1

Reviewer 1 Report

This study assesses the use of mandibular advancement devices in patients with obstructive sleep apnea with low threshold arousals.

1. The main problem and limitation of the present investigation is the use of a home sleep study without EEG monitoring during sleep instead of polysomnography.

2. References must be numbered in order of appearance in the text (and listed individually at the end of the manuscript.

Author Response

Point 1.  The main problem and limitation of the present investigation is the use of a home sleep study without EEG monitoring during sleep instead of polysomnography

Response 1: We thank the reviewer for the comment. The most important limitation of our study is the failure to use EEG. However, in our study, we performed manual scoring of home sleep tests and tried to understand the most relevant pathophysiological traits of the patients. Studies by Eckert and other experts on sleep breathing disorders suggest that expert scoring of polygraph tests could help to identify the prevalent pathophysiological characteristics of patients. Recently, anatomic and physiologic phenotypes have been identified as critical variables for choosing and predicting the success of OSA treatment and include both anatomic and functional factors, such as critical pressure for upper airway closure (Pcrit), arousal threshold, and muscle responsiveness, all of which play an essential role in the pathophysiology of OSA, so when certain pathophysiologic traits are prevalent, some therapies have a high probability of failure, but this in the advanced mandibular device is not well studied.  It is known in the literature that patients with high loop gain have a poor prognostic factor of advanced mandibular device efficacy. Thus, during the observation of home sleep tests of our patients, we noticed that patients with a lower response to therapy had the polygraphic features of a prevalent low arousal threshold. However, these polygraphic features were not measurable, so we sought to use Edwards' clinical predictor of low arousal threshold.

Point 2:   References must be numbered in order of appearance in the text (and listed individually at the end of the manuscript.

Response 2: I thank the reviewer for the comment. We have corrected the numbering of the references.

Reviewer 2 Report

This manuscript by Antaonaglia and collaborators deals with an application of a non-invasive technique (Edwards score) to determine the influence of a low arousal threshold (ArTH) on the response to an oral appliance in patients with an OA. This is in general an interesting approach to enable precision medicine in patients with this disorder. The manuscript in general is wordy and in part complex to follow. Linguistics may be improved. A shortening of the introduction and the discussion would considerably improve the structure and understanding of the study. The hypothesis stated was that a presence of a low ArTH predicts a poor response to OA or a treatment failure. Hence, the readability of this paper would also improve significantly if this hypothesis is adhered to and addressed repeatably in the manuscript.

General questions on design:

It appears as more than 5 years was used for data collection. Is it really correct that you recruited “consecutive patients” (see pg 2, l 95) during this period? Was the study design already determined and in print at study start in 1995? What were the conditions relating to the sleep studies during this extended period? Were they maintained or altered? Were the scores and the scorer the same? In particular hypopnea classification may be crucial with the technique used. Which criteria were applied? Did you attempt to apply alternative criteria? This might have altered classification considerably. Could there have been a period effect (repeated recordings) in the study? Also, please define the Edwards score better for readers that may not be familiar with this technique. Finally, did the characteristics of “consecutive” patients change during the long recruitment period? For instance, did you tend to receive referrals of patients who had already failed CPAP and thereby loosing the representativeness of you population.

A number of minor details may be addressed as they influence the stringency. Use either MAD or OA to describe the intervention. You currently invariably use both terms which is confusing. The conclusion in the abstract addresses the hypothesis. However, there is no data sited in the Results section (of the abstract) that support the conclusion. Please reformulate. In the introduction you refer to a reduction of mortality rate (pg 2, line 60). The Guo metanalysis is cited after the wrong sentence in the text and there are a number of studies that do not support that statement. Perhaps widen the scope of references a bit to include also CV morbidity?

Is figure 1 needed? Perhaps a Supplement? What type of validation did you do with respect to repeated assessments? Did you do allow repeated adjustments and if so when and how? Did you use the same rater of the Mallampati score in all patients? How did you assure compliance? Did you apply some type of a minimum criteria to describe compliance? On page 6 you introduce a 4 mm definition of a potential predictor of a response. This is not defined in the methods,

Regarding statistics. Did you adopt a power calculation before the start of the study in order to determine how many patients that needed to be included? Was this study submitted to Clinical Trials.gov and registered as NCT04323235929? I was unfortunately unable to locate it under this heading.

This author suggests that the Result section is restructured to align with the hypothesis and to, in a step-by-step manner, lists components that may had influenced the association between a low ArTH and the response to OA intervention. First do the baseline findings and thereafter the results of the intervention and finally the potential predictors. Please avoid unsupported data statements as symptomatic statements such as pg 4, line 170. Present in a table. What do mean by “reduction of symptomatic apneas”? For instance, in the current version you

Figures 2 and 3 depict grouped data. Considering the discussion on definitions and thresholds it might be better to describe individual data? Perhaps a so-called Spaghetti plot with delta values for the different categories (baseline AHI severity/low ArTH Y/N) would provide further information?

Page 7, line 210. I guess this is not variation but rather simple “polygraphic variables before and after treatment. Same page, line 211 avoid “who could be defined as…”

Discussion pg 8, line 234. This reviewer suggests you start with a brief summary description of the findings. Please state a comment on a potential explanation to why responses were seen only in a moderate category. Same page line 236-and onwards to page 9, line 311. Please put your findings into the perspective of previous findings. Also, please shorten all text that is peripheral to your findings. This will simplify the reading of your paper.

Page 9, line 313-14. What do you mean by only one device? Also, check language in the discussion. There are several errors.

Author Response

Point 1. It appears as more than 5 years were used for data collection. Is it correct that you recruited “consecutive patients” (see pg 2, l 95) during this period? Was the study design already determined and in print at the study started in 1995? What were the conditions relating to sleep studies during this extended period? Were they maintained or altered? Were the scores and the scorer the same?

Response 1: We thank the reviewer for the comment. The study was conducted between July 2015 and December 2020, the patients took the home diagnostic sleep test between 2015 and 2019, not in the same period, so it is not correct to say "consecutive." We have corrected this in the text, you can see it on page 4 on line 167: “Between July 2015 and December 2020, 34 patients were evaluated for inclusion into the study”

We collected data at different times to select the right patients. We used the same score and marker. In particular, we rechecked all studies to measure the Edwards score, and if the study was initially assessed by another person, we rechecked the study to be sure of our results.

Point 2 In particular hypopnea classification may be crucial with the technique used. Which criteria were applied? Did you attempt to apply alternative criteria? This might have altered classification considerably.

Response 2: Thank you very much for your comment, which is very important for the validity of Edwards' score. You point out an interesting issue because there are also two different definitions of hypopnea. In the Edwards et all study, they refer to the definition of hypopnea as generally the definition in the "American Academy of Sleep Medicine Manual for the Scoring of Sleep and Associated Events of 2007" in which hypopnea has two different definitions. Hypopnea is defined as a 30% reduction in airflow, as measured by the flow signal of the nasal pressure transducer, with a concomitant 4% drop in oxygen saturation, but also, alternatively, as a 50% or greater decline in flow signal associated with a 3% drop in oxygen saturation and/or an EEG arousal lasting at least 3 s. In our score, we always use the same criteria: hypopnea was defined as a 50% decrease in oronasal flow, followed by a greater than 3% decrease in SatO2 as stated at the beginning of the study.  However, we also considered that some work suggests that expert manual scoring of polygraph tests could help to identify the prevalent pathophysiological traits of patients. Thus, when observing the home sleep tests of our patients, we noticed that patients with a lower response to therapy had the polygraph features of a prevalent low arousal threshold. However, these polygraphic features were not measurable, so we use Edwards' clinical predictor of low arousal threshold.

Point 3 Could there have been a period effect (repeated recordings) in the study? Also, please define the Edwards score better for readers that may not be familiar with this technique. Finally, did the characteristics of “consecutive” patients change during the long recruitment period? For instance, did you tend to receive referrals of patients who had already failed CPAP and thereby lost the representativeness of your population?

Response 3 Thanks for your comment. We have not thought of a period effect, but it is possible. We have explained more about the Edwards score in the text: you can see it on page 2 to lines 59-69

“Arth refers to the neuromuscular mechanical pressure present at the end of apnea-hypopnea event, responsible for awakening from sleep–arousal and can only be quantified invasively by an epiglottic or esophageal pressure catheter5. A recent study by Edwards et al. reported that low AT could be estimated non-invasively through the following clinical score which attributes a point for each criterion met between an AHI ≤ 30 events/h, a nadir SpO2 ≥ 82.5%, and a hypopnea fraction of total respiratory events of>58.3%. A score of ≥2 predicts a low AT in OSAS patients, with high sensitivity and specificity (80.4% and 88% respectively)6. Other authors support the idea that it is possible to identify which of the non-anatomical factors contributes most to the pathogenesis of OSAS based on the characteristics of the sleep study tracing.”

We didn't think patient characteristics changed much, we select our patients carefully and consecutively.  We enrolled some patients who refused CPAP, but also most of them who had obvious elements to be treated with MAD. All severe and 10 moderate patients had a trial with CPAP first (so only 18 patients) and refused this treatment, the others were selected by MAD as the first choice. 

Point 4 Several minor details may be addressed as they influence the stringency. Use either MAD or OA to describe the intervention. You currently invariably use both terms which is confusing. The conclusion in the abstract addresses the hypothesis. However, there is no data cited in the Results section (of the abstract) that support the conclusion. Please reformulate.

Response 4 thanks for the comment. We have checked our use of MAD or OA. We have corrected the results section of the abstract.

Point 5  In the introduction you refer to a reduction in mortality rate (pg 2, line 60). The Guo metanalysis is cited after the wrong sentence in the text and several studies do not support that statement. Perhaps widen the scope of references a bit to include also CV morbidity?

Response 5 Thank you for your comment. You are right because the literature is very controversial about the effect of CPAP on the cardiovascular endpoint. We have corrected it. You can see it on page 2 to lines 69-72: “To date the use of a Continuous Positive Airway Pressure (CPAP), through a nasal or oronasal mask, is the only treatment documented as being effective in suppressing respiratory disturbances during sleep and improving clinical manifestations.  Its use for more than six hours has decreased sleepiness, improved daily functioning, and restored memory to normal levels8 9

Point 6 Is figure 1 needed? Perhaps a Supplement?

Response 6  We have removed it.

Point 7 What type of validation did you do concerning repeated assessments? Did you allow repeated adjustments and if so when and how? Did you use the same rater of the Mallampati score in all patients? How did you assure compliance? Did you apply some type of minimum criteria to describe compliance? On page 6 you introduce a 4 mm definition of a potential predictor of response. This is not defined in the methods,

Response 7 Each patient received repeated adjustments for each visit (1, 2, and 6 months) and we used the same Mallampati score for each patient. We check compliance with a patient diary and patients without compliance were excluded from the study (but all patients were compliant, usually MAD compliance is lost later in our patients, however, we defined good compliance for CPAP when patients use the device more than 75% of nights, for more than 4 hours). We have written these data and mm definitions in the text. 

Point 8 Regarding statistics. Did you adopt a power calculation before the start of the study to determine how many patients that needed to be included? Was this study submitted to Clinical Trials.gov and registered as NCT04323235929? I was, unfortunately, unable to locate it under this heading.

Response 8 We didn’t calculate the patients who needed to be included before, also the number is an internal number of the hospital, we have deleted it.

Point 9. This author suggests that the Result section is restructured to align with the hypothesis and to, in a step-by-step manner, lists components that may have influenced the association between a low Arth and the response to OA intervention. First, do the baseline findings and thereafter the results of the intervention and finally the potential predictors. Please avoid unsupported data statements as symptomatic statements such as pg 4, line 170. Present in a table. What do mean by “reduction of symptomatic apneas”? For instance, in the current version you

Response 9. Thank you to the reviewer. We have rewritten this part.

Point 10 Figures 2 and 3 depict grouped data. Considering the discussion on definitions and thresholds it might be better to describe individual data. Perhaps a so-called Spaghetti plot with delta values for the different categories (baseline AHI severity/low ArTH Y/N) would provide further information?

Response 10 thank you to the reviewer for the suggestion. Figure 4 has been added to show individual patient data and mean AHI variation both in Low Arth and non-Low ArTH groups

Point 11 Page 7, line 210. I guess this is not variation but rather simple “polygraphic variables before and after treatment. Same page, line 211 avoids “who could be defined as…”

Response 11 thank you. We have corrected it

Point 12  Discussion pg 8, line 234. This reviewer suggests you start with a summary description of the findings. Please state a comment on a potential explanation as to why responses were seen only in a moderate category. Same page line 236 and onwards to page 9, line 311. Please put your findings into the perspective of previous findings. Also, please shorten all text that is peripheral to your findings. This will simplify the reading of your paper. Page 9, lines 313-14. What do you mean by only one device? Also, check the language in the discussion. There are several errors.

Response 12  thank you. We have made the corrections you suggested.

Round 2
